# Enhancing Biomedical Lay Summarisation with External Knowledge Graphs

**Tomas Goldsack**[1], **Zhihao Zhang**[2], **Chen Tang**[3], **Carolina Scarton**[1], **Chenghua Lin**[1,4*]

[1]Department of Computer Science, University of Sheffield, UK
[2]College of Economics and Management, Beijing University of Technology, China,
[3]Department of Computer Science, The University of Surrey, UK
[4]Department of Computer Science, The University of Manchester, UK

{tgoldsack1, c.scarton}@sheffield.ac.uk  zhhzhang@bjut.edu.cn
chen.tang@surrey.ac.uk  chenghua.lin@manchester.ac.uk

## Abstract

Previous approaches for automatic lay summarisation are exclusively reliant on the source article that, given it is written for a technical audience (e.g., researchers), is unlikely to explicitly define all technical concepts or state all of the background information that is relevant for a lay audience. We address this issue by augmenting eLife, an existing biomedical lay summarisation dataset, with article-specific knowledge graphs, each containing detailed information on relevant biomedical concepts. Using both automatic and human evaluations, we systematically investigate the effectiveness of three different approaches for incorporating knowledge graphs within lay summarisation models, with each method targeting a distinct area of the encoder-decoder model architecture. Our results confirm that integrating graph-based domain knowledge can significantly benefit lay summarisation by substantially increasing the readability of generated text and improving the explanation of technical concepts.[1]

## 1 Introduction

Lay summarisation consists of generating a concise summary that illustrates the significance of a longer technical (or otherwise specialist) text and is comprehensible to the non-expert (Kuehne and Olden, 2015). A lay summary should contain minimal jargon and technical details (e.g., methodology), instead focusing largely on the simplification of key technical concepts and the explanation or relevant background information, thus allowing readers without technical knowledge to grasp the general topic and main ideas of an article (Srikanth and Li, 2021; Goldsack et al., 2022). However, since the original article is intended for a technical audience who already possess some domain knowledge, it

---

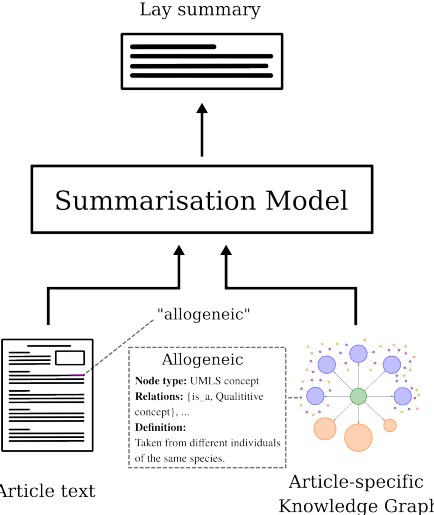

Figure 1: Overview of the "knowledge graph-enhanced Lay Summarisation" task formulation, exemplifying graph-based external information.

is unlikely to explicitly include all the information necessary for the lay summary, such as background details or definitions. As a result, lay summaries are often highly abstractive, adopting a simpler lexicon than the original article (Goldsack et al., 2022), and are typically written by experts who possess the knowledge required to simplify and explain the contents of the article (King et al., 2017).

Despite this disparity between lay summary and source article, automatic approaches to lay summarisation have typically relied solely upon the source article as input (Chandrasekaran et al., 2020; Guo et al., 2021; Luo et al., 2022a). Aiming to address this, we propose enhancing lay summarisation models with *external domain knowledge*, conducting the first study on **knowledge graph-enhanced lay summarisation** with a focus on biomedical articles. We augment eLife (Goldsack et al., 2022), an existing high-quality lay summarisation dataset, with article-specific knowledge graphs containing information on the technical concepts covered within the articles and the relation-

ships between them (exemplified in Figure 1), thus providing a structured representation of the domain knowledge that an expert human author might draw upon when writing a lay summary (§3). In doing this, we hypothesise that a model's ability to simplify and explain technical concepts for a lay audience will improve.

Although other forms of summarisation task (e.g., news articles) have seen significant research in the enhancing models using knowledge graphs (Huang et al., 2020; Zhu et al., 2021a; Lyu et al., 2022), this has yet to be explored for lay summarisation. To our knowledge, there is no work on determining the most effective way to incorporate graph-based knowledge for lay summarisation or other summarisation tasks. Therefore, we systematically investigate the effectiveness of **three different methods for injecting graph-based information into lay summarisation models** (§4) assessing them with both automatic and human evaluation (§5 & §6). Our results demonstrate that the integration of graph-based domain knowledge can significantly improve automatic lay summarisation enabling models to generate substantially more readable text and to better explain technical concepts.

## 2 Related Work

### 2.1 Lay Summarisation

The task of lay summarisation is a relatively novel one, introduced by the LaySumm subtask of the CL-SciSumm 2020 shared task series (Chandrasekaran et al., 2020). Introducing a multi-domain corpus of 572 article-lay summary pairs, the task attracted a total of 8 participants. The winning system, proposed by Kim (2020), adopted a hybrid approach, using a PEGASUS-based (Zhang et al., 2020) model to generate an initial abstractive lay summary before augmenting this with sufficiently readable article sentences extracted by a BERT-based model (Devlin et al., 2019).

Subsequent lay summarisation work has focused almost exclusively on the introduction and benchmarking of new corpora (all from the biomedical domain), rather than introducing specific modelling approaches for the task. Guo et al. (2021) introduce CDSR, a dataset derived from the Cochrane Database of Systematic Reviews, whereas Goldsack et al. (2022) introduce PLOS and eLife, two datasets derived from different biomedical journals (the Public Library of Science and eLife

journals, respectively).[2] Both studies benchmark their datasets with widely-used summarisation approaches, with BART variants (Lewis et al., 2020) invariably achieving the strongest performance. In another highly related work, Luo et al. (2022a) address the task of readability-controlled summarisation using data derived from PLOS, training a BART-based model to produce both the abstract and lay summary of an article in a controlled setting.

In contrast to previous works, we investigate the unexplored approach of modelling and integrating structured domain knowledge into lay summarisation models using article-specific knowledge graphs.

### 2.2 Knowledge Graph-Enhanced Text Generation

In recent years, the utilisation of knowledge graphs (KGs) containing external knowledge for text generation has seen increased interest, particularly when it comes to the modelling of commonsense knowledge. In particular, works focusing on tasks such as dialogue generation (Zhou et al., 2018; Tang et al., 2023), commonsense-reasoning (Liu et al., 2021), story generation (Guan et al., 2019; Tang et al., 2022), essay generation (Yang et al., 2019) have all seen the introduction of commonsense KG-enhanced models.

Some recent work has also focused on using KGs for abstractive summarisation, but tending towards modeling internal knowledge. Aiming to improve the faithfulness and informativeness of summaries, Huang et al. (2020), Zhu et al. (2021a), and Lyu et al. (2022) all utilise OpenIE to construct fact-based knowledge graphs from source documents (news articles). Huang et al. (2020) and Zhu et al. (2021a) extract graph node features using graph attention networks (Veličković et al., 2017), before incorporating these into the summarisation model decoder using an attention mechanism. Lyu et al. (2022) instead make use of additional semantic loss measures to attempt to capture extracted facts using an adapted pointer-generator network.

In contrast to previous works, we apply KG-based techniques to biomedical lay summarisation (as opposed to news article summarisation), a domain with additional challenges, including the extensive length of input articles and the presence of

---

[2] A version of these datasets with different test sets is also used within the BioLaySumm 2023 shared task (Goldsack et al., 2023a), that ran in parallel with this work.

complex technical concepts that need to be simplified or explained. Furthermore, due to the unique requirements of the task, we innovate by constructing knowledge graphs largely using *external* domain knowledge sources rather than from the article itself, as is the case with previous approaches.

## 3 Article Knowledge Graphs

We augment eLife, an existing dataset for biomedical lay summarisation with heterogeneous article-specific knowledge graphs (KGs). Each KG contains structured information on the complex biomedical concepts covered within the article and the relationships between them. In order to localise this information and provide an indication of where in an article a concept is mentioned, we also choose to model the article's section-based document structure within our graphs through the use of section-specific nodes. In the following, we describe in detail: 1) our process for extracting the knowledge that is used by our model (§3.1), and 2) how we structure that knowledge within a knowledge graph (§3.2). The methods by which we integrate graph-based knowledge into summarisation models are discussed in §4.

### 3.1 Knowledge Extraction

To extract relevant domain knowledge for an article, we draw upon the Unified Medical Language System (UMLS) (Bodenreider, 2004). This rich and actively-maintained resource has long been used as a key knowledge source for NLP in the biomedical domain (McCray et al., 2001; Demner-Fushman et al., 2010; Kang et al., 2021) and is comprised of three primary components: the Metathesaurus, the Semantic Network, and the Specialist Lexicon and Lexical Tools. The Metathesaurus is an extensive multi-lingual vocabulary database containing information on a large number of biomedical concepts, including their various names and definitions. The Semantic Network defines a set of *semantic types* that represent broad subject categories into which *all* concepts in the Metathesaurus can be assigned. Additionally, high-level relationships that occur between different semantic types are also defined. To extract the UMLS concepts mentioned within a given article, we utilise MetaMap (Aronson and Lang, 2010), one of the Lexical Tools provided alongside UMLS for this exact purpose, that is widely used in previous work (Sang et al., 2018; Sharma et al., 2019; Lai et al., 2021). For all arti-

| Metric | Abstract | Lay Summary | Definitions |
|---|---|---|---|
| FKGL↓ | 15.57 | 10.92 | 10.55 |
| CLI↓ | 17.68 | 12.51 | 13.02 |
| DCRS↓ | 11.78 | 8.83 | 10.36 |
| WordRank↓ | 9.21 | 8.68 | 8.6 |

Table 1: Mean readability scores for abstracts, lay summaries, and key UMLS definitions for eLife. FKGL = Flesch-Kinkaid Grade Level, CLI = Coleman-Liau Index, DCRS = Dale-Chall Readability Score.

cles in eLife, we apply MetaMap to each section in turn, retrieving all mentioned UMLS concepts. We restrict MetaMap to only a select number of English vocabularies without prohibitive access restrictions, but otherwise run it using default settings.

In line with observations made in previous works (Lai et al., 2021), we found that MetaMap, whilst succeeding in linking biomedical entities mentioned in the text with their corresponding UMLS concepts, also frequently returned a number of irrelevant concepts. Therefore, we adopt a text overlap-based approach to filter the original pool of extracted concepts for a given section, which we empirically found to eliminate the vast majority of the noise.[3]

For each remaining UMLS concept, we retrieve all semantic types with which it is associated, in addition to the formal definitions of both the concept and semantic types. Notably, these definitions are used as an integral component within all three KG-enhancement methods.[4] In order to confirm their suitability for a lay audience, we calculate and compare their average readability scores with those reported by Goldsack et al. (2022) for both the technical abstracts and lay summaries of eLife articles. The results of this analysis, given in Table 1, show that the UMLS definitions obtain scores that are overall much closer to those of the lay summaries than the abstracts, actually exceeding them in two out of the four metrics (FKGL and WordRank). An example of the definition format used for text augmentation is given in Figure 6 in the Appendix. In the next section, we describe how we represent all extracted information within article-specific knowledge graphs.

---

[3]More details on MetaMap vocabularies, noise reduction process, and the average article KG statistics are provided in the Appendix.

[4]Concepts without a formal UMLS definition are also removed from the final pool.

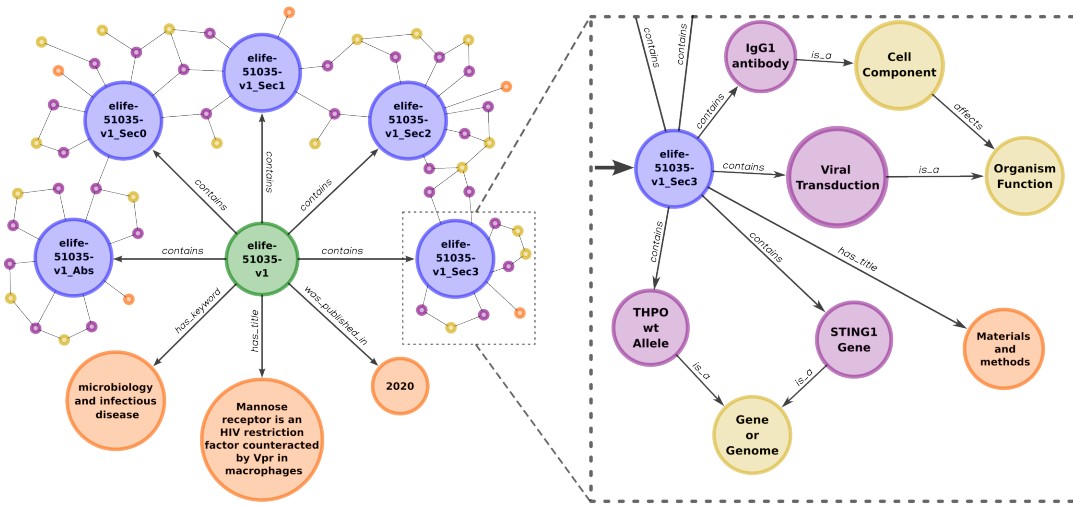

Figure 2: Example of article knowledge graph structure. Graph nodes are coloured as follows: Document, Section, Metadata, Concept, Semantic type.

## 3.2 Graph Construction

Each graph $\mathcal{G} = \{\mathcal{V}, \mathcal{E}\}$, where $\mathcal{V}$ is a set of nodes (or entities) and $\mathcal{E}$ is a set of edges. Each edge $e_{ij} \in \mathcal{E}$ defines a relation $r_{ij}$ between entities $v_i, v_j \in \mathcal{V}$, and thus can be represented as a triplet $e_{ij} = (v_i, r_{ij}, v_j)$. All graphs are heterogeneous, containing multiple types of entities and relations. Figure 2 presents a visualisation of an article knowledge graph.[5] Each type of node is described below:

- **Document node** – the central *root node*, which is the ancestor of all other nodes in the graph. We label this node simply with the unique ID assigned to each article.

- **Section node** – each section node represents a specific titled section (e.g. Introduction) of the document, including the abstract. To label these nodes, we concatenate the article ID with "_Abs" for the abstract or "_Sec{i}" for other sections, where {i} is the index of the section (zero-based).

- **Metadata node** – identify additional information relating to the article or its specific sections. This includes article and section titles, article keywords, and the date of publication.

- **Concept node** – nodes representing UMLS concepts. These are labelled with their unique UMLS identifier (CUI).

- **Semantic type node** – nodes representing semantic types from the Semantic Network. These are labelled with their unique Semantic Type identifier (TUI).

In addition to the 54 different relationship types defined within the semantic network (e.g., *affects* in Figure 2), we define several relations in order to represent the graph structure and additional metadata. Specifically, we define the relations *contains*, *was_published_in*, *has_title*, and *has_keyword*.

## 4 Knowledge-Enhanced Lay Summarisation Approaches

We investigate the effectiveness of three different methods for incorporating external knowledge from article graphs into encoder-decoder-based summarisation models. Our experiments are carefully designed so as to target a distinct aspect within the model architecture (i.e., the input, the encoder, and the decoder) with each selected method, taking inspiration from models that have recently been proven effective in the domain of news article summarisation (Zhu et al., 2021a; Pasunuru et al., 2021). Figure 3 provides a visualization of how each of these approaches fits into this architecture. To allow the ingestion of the full input article, we make use of Longformer Encoder-Decoder (Beltagy et al., 2020) as our base model for all experiments. This BART-based model replaces standard transformer self-attention with a sparse attention mechanism that scales linearly to the sequence length, enabling the processing of longer

---

[5]Note that, for visual clarity, this example contains significantly fewer concept and semantic type nodes than are present in the actual article graph.

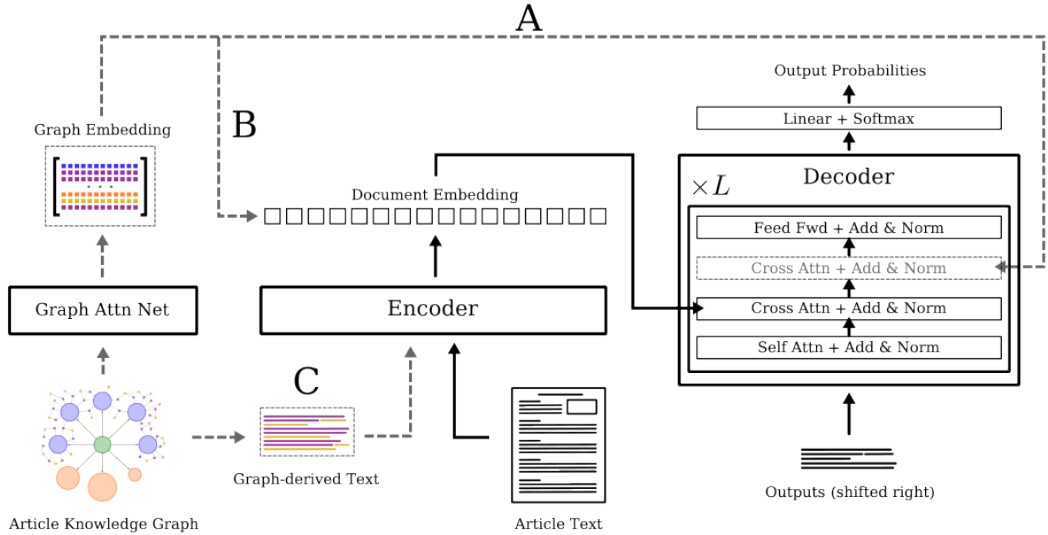

Figure 3: Visualisation of how our various knowledge-enhancement approaches incorporate external knowledge from article knowledge graphs into a transformer-based encoder-decoder architecture (as described in §4). **A**) Decoder cross-attention, **B**) Document embedding enhancement, **C**) Article text augmentation.

texts (such as research articles). We describe each knowledge-enhancement approach in detail below.

**(A) Decoder cross-attention.** We make use of a Graph Attention Network (GAT) (Veličković et al., 2017) to obtain an embedding of the article graph $\mathcal{G}$ in parallel with the base model encoder.

$$H^{\mathcal{G}} = \text{GAT}(\mathcal{G}) \qquad (1)$$

These Graph Neural Network (GNN) models produce a final set of node features (i.e., a graph embedding) by using attention layers to efficiently aggregate over the features of neighboring nodes, and are widely used in current literature to aggregate graph-based information for NLG tasks (Huang et al., 2020; Zhu et al., 2021b; Liu et al., 2021). During the decoding phase, we follow previous works (Zhu et al., 2021a) by forcing our model to attend to the KG embedding $H^{\mathcal{G}}$. Specifically, in every transformer layer of the decoder, we include a second cross-attention mechanism that occurs directly after the standard encoder cross-attention (see arrow A in Figure 3) and attends to the output of the GAT-based model.

**(B) Document embedding enhancement.** Again, we obtain an embedded graph representation $H^{\mathcal{G}}$ using the GAT model, but rather than attending to graph embeddings during decoding, we follow Pasunuru et al. (2021), combining the embedded node information into the final document embedding (i.e., the output of the encoder).

Specifically, we concatenate the document and graph embeddings, before passing them through an additional encoder layer. For a given input document $X$, this process can be formalised as follows:

$$H^X = \text{Encoder}(X) \qquad (2)$$

$$H^C = [H^X; H^{\mathcal{G}}] \qquad (3)$$

$$H^* = p \cdot \text{EncoderLayer}(H^C) + (1-p) \cdot H^C \quad (4)$$

where $H^X$ is an embedding of document $X$, $H^C$ is the concatenated document and graph embeddings, $H^*$ is the final 'enhanced' document embedding that is subsequently attended to during decoding, and $p$ is a scaling factor controlling the extent to which the additional encoder layer output is incorporated in the final enhanced document embedding. Note that $p$ is treated as a hyperparameter to the model, for which a value of 0.25 was found to provide the strongest validation set performance.[6]

**(C) Article text augmentation.** We also experiment with simply augmenting the input text with textual explanations of the key concepts (and their relations) derived from the graph. Whilst this may be arguably the most 'natural' way for a PLM-based model to interpret external information, this approach leads to an exponential increase in the

---

[6]We also find that a large value of $p$ causes significant degradation in performance, suggesting that the original document information is lost.

number of tokens required to describe each element, hence it is restricted to a set of few concepts. We select only those entities which are likely to be most central to the topic of the article (and, therefore, relevant to the lay summary). Specifically, we take the concept nodes that are mentioned in the article abstract and use the graph relations and retrieved definitions to provide a textual explanation of these salient concepts and their semantic types, which is then prepended to the article text. This takes the format of "*{concept_name} = {concept_definition}. {concept_name} is a {semtype_name}.*" repeated for each selected concept, followed by semantic type definitions formatted "*{semtype_name} = {semtype_definition}*" repeated for all mentioned semantic types.

## 5   Experimental Setup

### 5.1   Data

We derive knowledge graphs for all articles in eLife (Goldsack et al., 2022), a dataset for biomedical lay summarisation containing 4,828 article-summary pairs. Target summaries are expert-written lay summaries (i.e., summaries with a non-expert target audience) and inputs are the full text of the corresponding biomedical research articles. As explained in §1, we believe this task to be particularly suitable for domain knowledge augmentation due to the contrast in the level of expertise of the target audience between source and target which causes a discrepancy in the language used (specifically, reducing or explaining jargon terms) and the level of background information required.[7]

### 5.2   Baselines

As a baseline model, we include BART (Lewis et al., 2020), the state-of-the-art benchmark reported by Goldsack et al. (2022) for eLife, as well as in other previous lay summarisation works (Guo et al., 2021). Additionally, we include the reported performance of BART$_{scaffold}$ (Goldsack et al., 2022), a variant of BART trained to produce both the abstract and lay summary of an article in a controlled setting, which is equivalent to the model proposed by Luo et al. (2022a).[8]

---

[7]This discrepancy is evidenced in the analysis provided by Goldsack et al. (2022).

[8]Note that the original code for Luo et al. (2022a) is not yet available at the time of writing and their results are reported on a different dataset and thus are not comparable.

### 5.3   Implementation and Training

Each knowledge-based approach is implemented by manually adapting the Longformer implementation from Huggingface (Wolf et al., 2020) and, following previous work on lay summarisation (Chandrasekaran et al., 2020; Luo et al., 2022b; Goldsack et al., 2022), uses the full article text as input. For GAT-based models, we make use of the Deep Graph Library package (Wang et al., 2019) to implement a 3-layer GAT with 4 attention heads at each layer. For article graphs, we vary our node initialisation approach based on node type (as defined in §3.2). Specifically, we initialise concept and semantic type node features, with the embeddings of their textual definitions; document and section nodes with the embeddings of their title text (with title metadata nodes being subsequently ignored); and remaining metadata nodes (publication date and keywords) with embeddings on their textual content. All embeddings are generated using SciBert (Beltagy et al., 2019), a language model specifically trained on research papers from Semantic Scholar (Ammar et al., 2018) that is widely used for scientific data (Cohan et al., 2019; Cai et al., 2022; Goldsack et al., 2023b). Furthermore, all node embedding features are concatenated with one-hot features according to node type, as well as Random Walk Positional Encodings (Dwivedi et al., 2021). Following initialisation with `allenai/led-base-16384` checkpoint, we train all models on A100 GPUs and retain the checkpoint with the best validation set performance (more details provided in the Appendix).

### 5.4   Evaluation Setup

We conduct both automatic and human evaluations to provide a comprehensive assessment of how each knowledge-enhancement method affects the overall performance.

**Automatic evaluation.**   For each model, we report the average scores of several automatic metrics on the test split of eLife. As is common practice, we report widely-used summarisation metrics: BERTScore (Zhang et al., 2019) and the F1-scores of ROUGE-1, 2, and L (Lin, 2004).

To assess the readability of generated summaries, we report Flesh-Kincaid Grade Level (FKGL) and Dale-Chall Readability Score (DCRS), both of which compute an estimate of the US-grade level

| Model | Relevance | | | | Readability | | Factuality |
|---|---|---|---|---|---|---|---|
| | **R-1**↑ | **R-2**↑ | **R-L**↑ | **BeS**↑ | **CLI**↓ | **DCRS**↓ | **BaS**↑ |
| BART | 46.57 | 11.65 | 43.70 | 84.94 | 11.7 | 9.36 | -2.39 |
| BART$_{scaffold}$ | 45.28 | 10.99 | 42.51 | 84.65 | 11.32 | 9.19 | -2.57 |
| Longformer | 47.23 | 13.20 | 44.44 | 85.11 | 11.72 | 9.09 | -2.56 |
| – text-aug | **48.58*** | 14.24* | **45.71*** | **85.4*** | 10.94* | 8.72* | -2.45* |
| – doc-enhance | 48.10* | **14.43*** | 45.52* | 85.33* | **10.72*** | **8.50*** | **-2.35*** |
| – decoder-attn | 48.30* | 13.93 | 45.45* | 85.39* | 10.99* | 8.75* | -2.48* |

Table 2: Average performance of models on eLife test split. **R** = ROUGE F1, **BeS** = BERTScore F1, **CLI** = Coleman-Liau Index, **BaS** = BARTScore. * denotes that KG-enhanced model results are statistically significant with respect to the base model (Longformer) by way of Mann-Whitney U test.

required to comprehend a text.[9]

Additionally, we evaluate factuality using BARTScore (Yuan et al., 2021), which has been shown to have a strong alignment with human judgments of factual consistency in a recent study focusing specifically on long documents (Yee Koh et al., 2022). Following Yee Koh et al. (2022), we adapt BARTScore to use Longformer (thus allowing it to process the entire document as input) and fine-tune it on eLife.

**Human evaluation.** To provide a comprehensive assessment of the summaries generated by each knowledge-enhanced model, we conduct a human evaluation focusing on readability and factuality. Specifically, making use of 5 randomly sampled articles from the eLife test set, we ask human judges to evaluate each sentence within a generated summary along the following binary criteria: 1) *Factuality* - is the sentence factually correct (with respect to the source article); and 2) *Readability* - would a layperson be able to understand this sentence.[10] To help determine the factuality of the sentence, the annotator has access to the PDF of the source article as well as the reference lay summary.[11]

## 6 Experimental Results

### 6.1 Automatic Evaluation

Table 2 presents the performance of different models using the described automatic evaluation metrics on the test set of eLife. In addition to applying KG-enhancement methods in isolation, we also experiment with combining different methods, which

---

[9]Computed using the `textstat` package.
[10]An average of 68.5 sentences evaluated per model.
[11]Following Yee Koh et al. (2022), we encourage judges to use text-based search within the article to quickly identify relevant passages rather than asking them to read each article in full, reducing the cognitive burden placed upon them.

we largely find to be detrimental to model performance. Discussion and results (Table 6) of combined methods are provided in the Appendix. We discuss the performance of individually applied methods below, focusing on each aspect of automatic evaluation in turn.

**Relevance** Longformer can be seen to outperform the standard BART model in terms of relevance metrics, indicating that processing the entire document provides some benefit for lay summarisation. Additionally, all three knowledge enhancement methods significantly obtain improved scores across almost all relevance metrics (with the exception of R2 for the 'decoder attention' model). This provides a strong indication that the addition of graph-based domain knowledge provides models with relevant external information, enabling them to produce lay summaries that are closer in resemblance to the high-quality references.

**Readability** For readability metrics, it can first be noted that Longformer-based models obtain lower CLI and DCRS scores than those BART-based models. The calculation of CLI is based on the number of characters, words, and sentences it contains, whereas DCRS is based on the frequency of "familiar" (i.e., commonly-used) words, suggesting that Longformer produces summaries that are less syntactically and lexically complex.

We observe that the application of all knowledge enhancement methods results in improved scores for both metrics, with the document enhancement approach achieving the largest gains. This indicates that all knowledge enhancement methods are able to successfully influence the phrasing and structure of the summaries being generated by increasing the usage of more common (i.e., less technical) terminology. As reported in Table 1, the average CLI and DCRS scores for the reference lay summaries

| Model | # | Readability | Factuality |
|---|---|---|---|
| Longformer | 73 | 78.08 | 60.96 |
| – text-aug | 65 | 96.92* | 68.46 |
| – doc-enhance | 67 | 97.01* | 55.97 |
| – decoder-attn | 69 | 95.65* | 63.77 |

Table 3: Average percentage of generated sentences positively classified by judges for each high-level binary criteria. # = total number of sentences generated across all summaries. * denotes that KG-enhanced model results are statistically significant with respect to the base model (Longformer) by way of Mann-Whitney U test.

of eLife are 12.51 and 8.83, respectively.

**Factuality** For BARTScore (BAS), we again see a statistically significant improvement over the base Longformer model for all KG-enhancement methods, with the greatest improvement being obtained by the doc-enhance method. In order to gain further insight in these results, we also calculate the BARTScore values obtained by reference summaries, getting a mean score -2.39, which is similar to that of all tested models (and identical to that of BART). This suggests that all models are able to produce summaries with generative probabilities similar to that of the reference summaries. However, given that one model (doc-enhance) actually outscores the reference summaries, further analysis is needed to gain an understanding of the difference in the factual correctness of summaries produced by each method, for which we turn to our human evaluation.

### 6.2 Human Evaluation

Given the challenging and time-consuming nature of evaluating the factuality of technical biomedical sentences against the source article, we carefully plan our human evaluation so as to ensure reliability in our results. We employed two annotators to evaluate generated sentences following the procedure laid out in §5, both of whom are experts in NLP and familiar with common model shortcomings (e.g., hallucinations). Table 3 presents the total percentage of sentences that were positively classified for both readability and factuality averaged across evaluators, who achieve a Cohan's $\kappa$ of 0.42.

**Discussion** The results in Table 3 suggest that the application of all KG-enhancement methods causes a notable increase in the readability of the text produced by the model, with all models scor-

**a. [Meiosis]**
**Longformer** - During meiosis, the DNA in one of the chromosomes is copied and then the two copies are recombined so that each new generation will have a single copy of the gene that encodes the protein encoded by that gene. [1/2]
**w/ text-aug** - ... a process known as meiosis ... two copies of each chromosome are then exchanged between the newly formed cells, which results in a unique set of genes being passed on to the next generation. [2/2]

**b. [Glabrous skin / Mechanoreceptors]**
**Longformer** - The orientation of an object depends largely on how its edges activate mechanoreceptors in the glabrous skin of the fingertips. [0/2]
**w/ doc-enhance** - The fingertip's surface is covered by a ... layer of skin known as the glabrous skin. [2/2] These cells are responsible for sensing touch, and they are also responsible for detecting the orientation of objects that touch them. [2/2]

**c. [Slow wave sleep]**
**Longformer** - Most studies of sleep have focused on ... slow wave sleep, in which the brain's activity alternates between periods of alternating periods of slow and fast sleep. [0/2]
**w/ decoder-attn** - Slow wave sleep is characterized by rhythmic waves of electrical activity in the brain, which are thought to be part of the process by which the brain consolidates memories. [2/2]

Figure 4: A case study comparing how the application of each method affects the explanation of specific technical concepts within the human evaluation sample. Colours and superscript are used to denote the number of evaluators who judged the sentence as readable for a lay audience (e.g., [2/2] = 2 out of 2 evaluators).

ing significantly higher than the base Longformer model. Alternatively, the results for factuality show that, although there is a slight variance in performance between KG methods, none of them are judged to be by a statistically significant margin. These results indicate that all methods are able to effectively introduce relevant external information into the model, enabling it to produce text that is easier for a lay audience to comprehend without significantly compromising the factual correctness of the base model.

**Case Study** To gain a better insight into how knowledge enhancement methods influence the readability of generated summaries, we present a case study in Figure 4 in which we compare the explanations of specific technical concepts generated by KG-enhanced models and the base Longformer model, alongside their annotator ratings. [12]

These examples demonstrate how KG-enhancement methods improve the model's handling of technical concepts, thus making them easier to understand for a lay reader. Specifically,

---

[12]An extended version of this case study is given in Figure 5 in the Appendix.

examples show how methods can influence the model to generate an explanation in instances where the base model fails to provide one (b) or improve the explanation in instances where the base model's is difficult to understand (a and c).

## 7 Conclusion

This papers presents the first study on the use of knowledge graphs to enhance lay summarisation, augmenting the biomedical lay summarisation dataset eLife with article-specific knowledge graphs containing domain-specific external knowledge on relevant technical concepts. We compare three distinct approaches for incorporating graph-based knowledge into encoder-decoder summarisation models, placing an emphasis on the readability and factual correctness of the generated output. Our results suggest that integrating external knowledge has the potential to substantially improve lay summarisation, particularly for the generation of readable text and explanation of technical concepts. We would like to see future work investigate the use of additional graph representations, as well as their integration into larger models that adopt different architectures (e.g., decoder-only).

## Limitations

One possible limitation of our work is derived from the use of resources from UMLS (i.e., UMLS concept names, semantic types and relations, definitions, etc.). Accessing these resources requires an individual license with the US National Library of Medicine (NLM), and their subsequent distribution is restricted by this license agreement. Therefore, it is likely that we will have to confirm the license status of those who wish to have access to the knowledge-graph resources used in this work. In an attempt to reduce any potential impacts this will have on the ability to share our resources, we only make use of only a select number of vocabularies less restrictive licences. More details on the vocabularies used are provided in the Appendix.

## 8 Acknowledgements

This work was supported by the Centre for Doctoral Training in Speech and Language Technologies (SLT) and their Applications funded by UK Research and Innovation [grant number EP/S023062/1].

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

## A  Appendix

| Node type | Average Count |
|-----------|---------------|
| Document  | 1.00 |
| Section   | 5.90 |
| Metadata  | 7.49 |
| Concept   | 364.93 |
| SemType   | 63.47 |

Table 4: The average node type frequency statistics for a single article in the train split.

**Graph Statistics**  Table 4 presents the average node type frequencies in the graph of a given article. Additionally, Tables 7 and 8 present the average semantic type (SemType) node frequencies, and Table 9 presents the average relation frequencies (all of which are located at the end of the Appendix).

**MetaMap - UMLS vocabularies.**  As mentioned in §3, we restrict MetaMap to a select number of English vocabularies with access restrictions lower than level 4.[13]  Specifically, we allow MetaMap to use the following vocabularies: Alcohol and Other Drug Thesaurus (AOD), Diseases Database (DDB), CRISP Thesaurus (CSP), DrugBank (DRUGBANK), Diagnostic and Statistical Manual of Mental Disorders - Fifth Edition (DSM-5), Human Phenotype Ontology (HPO), NCI Thesaurus (NCI), MedlinePlus Health Topics (MEDLINEPLUS), LOINC (LNC), MeSH (MSH), RxNorm (RXNORM), and Gene Ontology (GO).

**MetaMap - Noise reduction.**  We found that MetaMap, in addition to accurately identifying UMLS concepts that were mentioned in a passage of text, often also returned a large number of unwanted concepts. These typically were only

---

[13] Information on all possible MetaMap vocabularies and their access restrictions can be found on the following page: https://www.nlm.nih.gov/research/umls/sourcereleasedocs/index.html

| Model | Abstractiveness (%) |
|-------|---------------------|
| Longformer | 22.95 |
| – text-aug | 20.99 |
| – doc-enhance | 20.00 |
| – decoder-attn | 24.46 |

Table 5: The average abstractiveness (measured in terms of novel 1-grams) of summaries generated using different graph integration methodologies.

distantly relevant to the text - for example, multi-word concepts with one matching word. Therefore, for each section of an article, we employed a simple word overlap-based to filter out unwanted retrieved concepts. Specifically, after removing stopwords from the main text and lemmatisation concept names, we retain only the concepts for which all words in its name were found to appear in the section text. We empirically found this approach to significantly outperform word embedding-based approaches, which often failed to filter out irrelevant multi-word concepts if a single word appeared in the main text.

**Additional implementation and training details.**
We employ Longformer Encoder Decoder (LED) with the `allenai/led-base-16384` Huggingface checkpoint as our base model for all knowledge enhancement approaches, using the default parameters for model training and an input limit of 8192.

For the "decoder cross-attention" method, we replicate the standard Huggingface LED multi-head decoder cross-attention implementation (where the head no. is determined by the model configuration), making use of the GAT-produced graph embedding as the key and value matrices (in place of the encoder output), with the query matrix being the output of the previous standard cross-attention module.

For the "document embedding enhancement" method, we replicate the standard LED encoder layer, adapting the configuration to account for the larger input of the concatenated document and graph feature representations.

For the "article text augmentation" method, we prepend the article text with the graph-derived text and shown in Figure 6, extending the input text limit to 16,384 to accommodate the lengthy augmentation text without losing article text. All models are trained for a maximum of 20 epochs, retaining the checkpoints that achieved the highest validation set performance (average ROUGE score across variants).

The time taken to train each model on 2 A100 GPUs ranged from 12 hours to 2 days depending on the specific methodology used, with the GAT-based methods - "document embedding enhancement" and "decoder cross attention" - taking longer than the "article text augmentation" method.

**Summary abstractiveness.**  In an effort to gain further insight into how each KG-enhancement

| Model | Relevance | | | | Readability | | Factuality |
|---|---|---|---|---|---|---|---|
| | R-1↑ | R-2↑ | R-L↑ | BeS↑ | CLI↓ | DCRS↓ | BaS↑ |
| Longformer | 47.23 | 13.20 | 44.44 | 85.11 | 11.72 | 9.09 | -2.56 |
| – text-aug + doc-enhance | 46.28 | 13.45 | 43.77 | 84.72 | 11.37 | 8.06 | -2.56 |
| – text-aug + decoder-attn | 47.27 | 13.93 | 44.52 | 84.95 | 10.47 | 8.15 | -2.57 |
| – doc-enhance + decoder-attn | 28.85 | 3.7 | 27.58 | 73.60 | 2.68 | 1.56 | -6.48 |

Table 6: Average performance of models with combinations of KG-enhancement methods on eLife test split. **R** = ROUGE F1, **BeS** = BERTScore F1, **CLI** = Coleman-Liau Index, **BaS** = BARTScore.

method influences summary generation, we measure the abstractiveness (as the percentage of novel 1-grams) of the generated summaries, providing the results in Table 5. Interestingly, the results show that both the "text augmentation" and "document enhancement" methods slightly decrease the abstractiveness of the base model, but the "decoder attention" method slightly increases it, suggesting that this method is more effective at introducing external vocabulary.

**Combining KG-enhancement Methods**    Table 6 presents the results obtained when applying different combinations of the KG-enhancement methods to Longformer. Interestingly, no combined model matches the overall performance of any single KG method model. Although these combinations achieve positive results for readability metrics, this generally comes at the expense of readability metrics, with the combination of text augmentation and decoder attention being the only model to achieve ROUGE scores equal to that of the base model.

We believe this performance degradation is likely a result of the dilution or loss of document information at the expense of graph-based information. This is similar to what was observed when the value of $p$, as given in equation (4), was set to too great a value for the document enhancement method. In the case of the method combinations that include the text augmentation method, the document information is likely more diluted as a result of the larger embedding size (as caused by the increased input size). Alternatively, when the document embedding and decoder attention methods are combined, the original document information seems to be lost, resulting in particularly high loss values during training and the model being unable to produce a coherent output (as evidenced by the extremely low scores for readability and factuality metrics).

**Case study**    Figure 5 presents an expanded version of the case study given in Figure 4, whereby we show all model outputs for each of the technical concepts. As in Figure 4, we see that the KG-enhanced models typically improve on the explanation proved by the base Longformer model.

> **[Meiosis]**
> **Longformer** - During meiosis, the DNA in one of the chromosomes is copied and then the two copies are recombined so that each new generation will have a single copy of the gene that encodes the protein encoded by that gene. [1/2]
> **w/ text-aug** - ... a process known as meiosis ... two copies of each chromosome are then exchanged between the newly formed cells, which results in a unique set of genes being passed on to the next generation. [2/2]
> **w/ doc-enhance** - ... meiosis depends on the rate and location of the recombination events, and on the length of the DNA that is involved in the process. [1/2]
> **w/ decoder-attn** - This is achieved through a process called meiosis, in which a cell divides to produce four daughter cells, each of which contains a copy of one of the four copies of the genes found on the parent cell's chromosomes. [2/2]
>
> **[Glabrous Skin / Mechanoreceptors]**
> **Longformer** - The orientation of an object depends largely on how its edges activate mechanoreceptors in the glabrous skin of the fingertips. [0/2]
> **w/ text-aug** - ... information about the orientation of an object is based largely on how its edges activate mechanoreceptors in the skin of the fingertips. [2/2] ... tactile neurons in the fingertips are able to detect the edges of objects with high sensitivity [2/2] ...
> **w/ doc-enhance** - The fingertip's surface is covered by a ... layer of skin known as the glabrous skin. [2/2] These cells are responsible for sensing touch, and they are also responsible for detecting the orientation of objects that touch them. [2/2]
> **w/ decoder-attn** - The orientation of an object is determined largely by how its edges contact the skin and activate nerve cells in the fingertips. [2/2]
>
> **[Slow Wave Sleep]**
> **Longformer** - Most studies of sleep have focused on a type of sleep known as slow wave sleep, in which the brain's activity alternates between periods of alternating periods of slow and fast sleep. [0/2]
> **w/ text-aug** - slow wave sleep is thought to help us to consolidate newly acquired memories and to consolidate our sense of awareness of the world around us. [2/2] ... in slow wave sleep, the propagation of slow waves is lost between the brain's regions. [2/2]
> **w/ doc-enhance** - Slow wave sleep is characterized by a slow oscillation of electrical activity in the brain. [2/2]
> **w/ decoder-attn** - Slow wave sleep is characterized by rhythmic waves of electrical activity in the brain, which are thought to be part of the process by which the brain consolidates memories. [2/2]

Figure 5: An extended case study comparing how the application of each method affects the explanation of specific technical concepts within the human evaluation sample. Colours and superscript are used to denote the number of evaluators who judged the sentence as readable for a lay audience (e.g., [2/2] = 2 out of 2 evaluators).

*{Concept definitions and relations}*

Alleles = Variant forms of the same gene, occupying the same locus on homologous CHROMOSOMES, and governing the variants in production of the same gene product. Alleles is a Gene or Genome.

Molecule = An aggregate of two or more atoms in a defined arrangement held together by chemical bonds. Molecule is a Substance.

Discover = See for the first time; identify. Discover is a Activity.

Histocompatibility = The degree of antigenic similarity between the tissues of different individuals, which determines the acceptance or rejection of allografts. Histocompatibility is a Qualitative Concept.

In Vivo = Located or occurring in the body. In Vivo is a Spatial Concept.

Species = A group of organisms that differ from all other groups of organisms and that are capable of breeding and producing fertile offspring. Species is a Classification.

Cells = The fundamental, structural, and functional units or subunits of living organisms. They are composed of CYTOPLASM containing various ORGANELLES and a CELL MEMBRANE boundary. Cells is a Cell.

Allogeneic = Taken from different individuals of the same species. Allogeneic is a Qualitative Concept.

Antigens = Substances that are recognized by the immune system and induce an immune reaction. Antigens is a Immunologic Factor.

Result = The result of an action. Result is a Functional Concept.

Major Histocompatibility Complex = The genetic region which contains the loci of genes which determine the structure of the serologically defined (SD) and lymphocyte-defined (LD) TRANSPLANTATION ANTIGENS, genes which control the structure of the IMMUNE RESPONSE-ASSOCIATED ANTI-GENS, HUMAN; the IMMUNE RESPONSE GENES which control the ability of an animal to respond immunologically to antigenic stimuli, and genes which determine the structure and/or level of the first four components of complement. Major Histocompatibility Complex is a Gene or Genome.

...

*{SemType definitions}*

Gene or Genome = A specific sequence, or in the case of the genome the complete sequence, of nucleotides along a molecule of DNA or RNA (in the case of some viruses) which represent the functional units of heredity.

Substance = A material with definite or fairly definite chemical composition.

Activity = An operation or series of operations that an organism or machine carries out or participates in.

Classification = A term or system of terms denoting an arrangement by class or category.

Cell = The fundamental structural and functional unit of living organisms.

Qualitative Concept = A concept which is an assessment of some quality, rather than a direct measurement.

Spatial Concept = A location, region, or space, generally having definite boundaries.

Immunologic Factor = A biologically active substance whose activities affect or play a role in the functioning of the immune system.

Functional Concept = A concept which is of interest because it pertains to the carrying out of a process or activity.

Quantitative Concept = A concept which involves the dimensions, quantity or capacity of something using some unit of measure, or which involves the quantitative comparison of entities.

Temporal Concept = A concept which pertains to time or duration.

...

*{Article text}*

Figure 6: Text augmentation example.

| Semantic type | Average Count |
|---|---|
| Manufactured Object | 0.98 |
| Classification | 0.61 |
| Qualitative Concept | 1.0 |
| Disease or Syndrome | 0.51 |
| Health Care Activity | 0.95 |
| Occupational Activity | 0.94 |
| Phenomenon or Process | 0.99 |
| Individual Behavior | 0.26 |
| Intellectual Product | 1.0 |
| Organism Attribute | 0.96 |
| Health Care Related Organization | 0.47 |
| Body Part, Organ, or Organ Component | 0.86 |
| Social Behavior | 0.67 |
| Therapeutic or Preventive Procedure | 0.78 |
| Research Device | 0.13 |
| Spatial Concept | 1.0 |
| Temporal Concept | 1.0 |
| Behavior | 0.45 |
| Environmental Effect of Humans | 0.05 |
| Mammal | 0.76 |
| Research Activity | 1.0 |
| Quantitative Concept | 1.0 |
| Occupation or Discipline | 0.81 |
| Functional Concept | 1.0 |
| Conceptual Entity | 0.81 |
| Activity | 1.0 |
| Population Group | 0.65 |
| Age Group | 0.35 |
| Mental Process | 0.92 |
| Natural Phenomenon or Process | 0.93 |
| Geographic Area | 0.78 |
| Biologic Function | 0.43 |
| Human-caused Phenomenon or Process | 0.20 |
| Idea or Concept | 1.0 |
| Body Location or Region | 0.58 |
| Finding | 1.0 |
| Organ or Tissue Function | 0.38 |
| Amino Acid, Peptide, or Protein | 0.89 |
| Injury or Poisoning | 0.13 |
| Professional or Occupational Group | 0.50 |
| Gene or Genome | 0.84 |
| Element, Ion, or Isotope | 0.91 |
| Body Substance | 0.56 |
| Cell | 0.91 |
| Receptor | 0.50 |
| Hazardous or Poisonous Substance | 0.80 |
| Pathologic Function | 0.59 |
| Organism | 0.46 |
| Indicator, Reagent, or Diagnostic Aid | 0.81 |
| Cell Component | 0.82 |
| Biologically Active Substance | 0.91 |
| Animal | 0.52 |
| Biomedical or Dental Material | 0.63 |
| Group | 0.38 |
| Body System | 0.26 |
| Physical Object | 0.34 |
| Antibiotic | 0.39 |
| Organism Function | 0.95 |
| Governmental or Regulatory Activity | 0.48 |
| Organization | 0.61 |
| Tissue | 0.40 |
| Diagnostic Procedure | 0.41 |
| Biomedical Occupation or Discipline | 0.68 |
| Entity | 0.40 |

Table 7: The average semantic node type frequency statistics for a single article in the train split.

| Semantic type | Average Count |
|---|---|
| Inorganic Chemical | 0.66 |
| Pharmacologic Substance | 0.93 |
| Physiologic Function | 0.33 |
| Immunologic Factor | 0.58 |
| Molecular Function | 0.67 |
| Clinical Attribute | 0.39 |
| Laboratory Procedure | 0.88 |
| Event | 0.59 |
| Human | 0.04 |
| Chemical Viewed Structurally | 0.50 |
| Sign or Symptom | 0.28 |
| Enzyme | 0.68 |
| Medical Device | 0.78 |
| Genetic Function | 0.73 |
| Nucleic Acid, Nucleoside, or Nucleotide | 0.75 |
| Organic Chemical | 0.88 |
| Patient or Disabled Group | 0.23 |
| Virus | 0.20 |
| Cell Function | 0.74 |
| Substance | 0.94 |
| Daily or Recreational Activity | 0.72 |
| Bacterium | 0.37 |
| Laboratory or Test Result | 0.17 |
| Neoplastic Process | 0.17 |
| Eukaryote | 0.33 |
| Amino Acid Sequence | 0.30 |
| Cell or Molecular Dysfunction | 0.57 |
| Regulation or Law | 0.09 |
| Chemical | 0.45 |
| Body Space or Junction | 0.23 |
| Nucleotide Sequence | 0.37 |
| Chemical Viewed Functionally | 0.37 |
| Family Group | 0.52 |
| Vertebrate | 0.18 |
| Fungus | 0.23 |
| Food | 0.40 |
| Embryonic Structure | 0.28 |
| Bird | 0.19 |
| Molecular Biology Research Technique | 0.56 |
| Molecular Sequence | 0.06 |
| Group Attribute | 0.03 |
| Vitamin | 0.16 |
| Mental or Behavioral Dysfunction | 0.30 |
| Hormone | 0.19 |
| Plant | 0.18 |
| Anatomical Structure | 0.06 |
| Fish | 0.14 |
| Machine Activity | 0.25 |
| Educational Activity | 0.09 |
| Reptile | 0.02 |
| Amphibian | 0.03 |
| Congenital Abnormality | 0.04 |
| Experimental Model of Disease | 0.06 |
| Archaeon | 0.02 |
| Language | 0.02 |
| Anatomical Abnormality | 0.01 |
| Acquired Abnormality | 0.01 |
| Fully Formed Anatomical Structure | 0.01 |
| Professional Society | 0.01 |
| Clinical Drug | 0.00 |
| Self-help or Relief Organization | 0.00 |

Table 8: The average semantic node type frequency statistics for a single article in the train split (continued).

| Relation type | Average Count |
|---|---|
| contains | 564.94 |
| has_keyword | 1.59 |
| has_title | 4.90 |
| was_published_in | 1.0 |
| is_a | 536.22 |
| branch_of | 0.86 |
| affects | 255.99 |
| performs | 25.23 |
| exhibits | 6.29 |
| conceptual_part_of | 10.80 |
| result_of | 134.99 |
| measures | 71.28 |
| issue_in | 97.14 |
| associated_with | 43.36 |
| occurs_in | 15.44 |
| connected_to | 2.61 |
| process_of | 101.00 |
| degree_of | 12.83 |
| manifestation_of | 37.81 |
| uses | 14.00 |
| location_of | 60.24 |
| causes | 42.52 |
| adjacent_to | 6.13 |
| tributary_of | 0.86 |
| prevents | 6.84 |
| produces | 67.90 |
| method_of | 13.61 |
| property_of | 6.45 |
| complicates | 54.91 |
| evaluation_of | 22.34 |
| co-occurs_with | 21.40 |
| carries_out | 8.23 |
| interacts_with | 102.22 |
| measurement_of | 17.90 |
| traverses | 1.45 |
| developmental_form_of | 3.52 |
| treats | 8.34 |
| surrounds | 4.74 |
| part_of | 27.68 |
| conceptually_related_to | 0.31 |
| precedes | 30.83 |
| consists_of | 6.54 |
| analyzes | 16.93 |
| assesses_effect_of | 23.76 |
| disrupts | 49.62 |
| contains | 2.97 |
| diagnoses | 8.38 |
| indicates | 3.30 |
| practices | 0.78 |
| manages | 0.77 |
| derivative_of | 0.28 |
| interconnects | 0.70 |
| ingredient_of | 0.05 |

Table 9: The average relation type frequency statistics for a single article in the train split.