# OpenReview forum: "Enhancing Biomedical Lay Summarisation with External Knowledge Graphs"
_EMNLP/2023/Conference — EMNLP 2023 Main_

### Official Review · Reviewer_cPxM · 2023-07-21

**Typos Grammar Style And Presentation Improvements:** Line 573
**Soundness:** 4

**Excitement:**

4: Strong: This paper deepens the understanding of some phenomenon or lowers the barriers to an existing research direction.

**Missing References:**

The related work section is comprehensive and I did not identify any important missing references.

**Paper Topic And Main Contributions:**

This paper presents an approach for lay summarisation of biomedical articles by incorporating information from knowledge graphs. The authors explore three different ways to inject the information from knowledge graphs in a base encoder-decoder architecture. The three approaches are independently tested and compared against state-of-the-art baselines, using the eLife dataset.


**Questions For The Authors:**

There are no specific questions to the authors.

**Reasons To Accept:**

A. The paper is well written, the explanations are detailed.

B. The three methods to inject knowledge graph information are intuitive and explore the impact of injecting such information in different stages of a base encoder-decoder architecture.

C. The evaluation compares the approach against well known and strong baselines, and the results show an improvement.

D. There is automatic evaluation using well-known evaluation metrics (ROUGE and BARTScore), plus an automatic assessment of readability. In addition, there is a human evaluation.

**Reasons To Reject:**

A. The base model is LongFormer. It is possible that the conclusions do not carry to other base models.

B. Only one dataset was used in the evaluation.

C. It is not stated whether the code will be available. This can make reproducibility more difficult, but this is compensated by the fact that the approach is explained in detail.


**Reproducibility:**

4: Could mostly reproduce the results, but there may be some variation because of sample variance or minor variations in their interpretation of the protocol or method.

**Reviewer Confidence:**

4: Quite sure. I tried to check the important points carefully. It's unlikely, though conceivable, that I missed something that should affect my ratings.

---

> ### Author Rebuttal · Authors · 2023-08-28
>
> Thank you for your insightful comments and your recognition of our work. Please find our response to specific comments below:
>
> __*The base model is LongFormer. It is possible that the conclusions do not carry to other base models.*__
>
> In Longformer, we selected what was the most appropriate model for the task, given its general popularity and its ability to process the entire input document (that has resulted in strong performance on a number of similar tasks). However, we agree that exploring additional base models is a useful direction for future work.
>
> __*Only one dataset was used in the evaluation.*__
>
> Although we agree that it would have been desirable to include more datasets for evaluation, we opted to use eLife exclusively due to the particularly high quality of its reference lay summaries (as described in [Goldsack et al. (2022)](https://aclanthology.org/2022.emnlp-main.724/)). Specifically, eLife’s lay summaries are written by journal editors who are experts in communitcating complex technical findings to a lay audience - something which is unique amongst lay summarisation dataset, of which only a few exist.
>
> __*It is not stated whether the code will be available. This can make reproducibility more difficult, but this is compensated by the fact that the approach is explained in detail.*__
>
> We can assure you that code will be made available and a GitHub URL included in a final paper version. This was simply not included in the current version so as to not break author anonymity.

---

### Official Review · Reviewer_6F5o · 2023-08-04

**Soundness:** 3

**Excitement:**

4: Strong: This paper deepens the understanding of some phenomenon or lowers the barriers to an existing research direction.

**Missing References:**

Integrating graph based knowledge for summarization:
https://link.springer.com/article/10.1007/s42979-023-01867-1 (this particular work considered contemporaneous, others not)
https://www.mdpi.com/2079-9292/9/9/1520
https://arxiv.org/abs/2204.13761
Any of the retrieval augmented language models, e.g. https://arxiv.org/abs/2005.11401

**Paper Topic And Main Contributions:**

The primary contribution of this paper is the application of external knowledge graphs (derived from UMLS and MetaMap) in lay summarization. Secondarily, the paper investigates three methods of incorporating the external knowledge graph information into an encoder/decoder model: (A) by concatenating a graph encoding of the document and external knowledge in the decoder, (B) by concatenating the graph and document encodings and interpolating between this and an additional layer, and (C) linearizing the graph representation and passing this into the transformer. It includes a thorough automatic evaluation and a human evaluation over a small set of documents, demonstrating the approach improves performance.

**Questions For The Authors:**

(A) Does the external knowledge graph decrease the abstractiveness of the summaries (target and generated)? In some sense adding the external vocabulary may be making the task more extractive than abstractive, which would be a contribution/finding worth including.

(B) What was the global attention setting?

(C) Why is it surprising that all methods of knowledge graph integration performed comparably? This is the same information, allowed to interact fairly richly with all the document tokens, and relatively minor differences in model architecture.

**Reasons To Accept:**

The paper presents a well-implemented NLP engineering experiment for lay-summarization of scientific articles.

**Reasons To Reject:**

(A) (arguably a presentation point) One of the paper’s main points is a lack of external knowledge in use while summarizing. LLMs _are_ a source of external knowledge while summarizing. This is a major theme in the paper, and diminishes the work.

(B) Graph Attention Networks are a single type of graph network. The paper documents different approaches to incorporating graph information into a transformer, not an exhaustive or even extensive search over graph representations, as claimed in the work.

(C) Manual evaluation was conducted over only 5 articles; while there may have been ~70 sentences evaluated, this does not give a particularly good measure for overall performance.

**Reproducibility:**

4: Could mostly reproduce the results, but there may be some variation because of sample variance or minor variations in their interpretation of the protocol or method.

**Reviewer Confidence:**

4: Quite sure. I tried to check the important points carefully. It's unlikely, though conceivable, that I missed something that should affect my ratings.

---

> ### Author Rebuttal · Authors · 2023-08-28
>
> Thank you for your insightful comments and your recognition of our work. Please find our response to specific comments and questions below:
>
> __*(arguably a presentation point) One of the paper’s main points is a lack of external knowledge in use while summarizing. LLMs are a source of external knowledge while summarizing. This is a major theme in the paper, and diminishes the work.*__
>
> We are sorry, but we do not really understand this comment, since this paper does not use LLMs. We agree that the use of LLMs also  has the potential to introduce external knowledge for Lay Summarisation, and this is  something that we would definitely like to explore in future work. However, in this work we do not make use of LLMs, and there are still many scenarios where a the use of LLMs may not be possible or preferable (e.g., in lower-resource settings), thus making the integration of external knowledge into smaller LMs (such a Longformer) a topic  still worthy  of investigation.
>
> __*Graph Attention Networks are a single type of graph network. The paper documents different approaches to incorporating graph information into a transformer, not an exhaustive or even extensive search over graph representations, as claimed in the work.*__
>
> We acknowledge that other types of graph network are also available and thank the reviewer for highlighting this. Our decision to make use of graph attention networks for our experiments was based on their popularity in the relevant literature (e.g., [Liu et al., 2021](https://ojs.aaai.org/index.php/AAAI/article/view/16796), [Huang et al. 2020](https://aclanthology.org/2020.acl-main.457/)) and, whilst we would also like to look into other types of graph representation in future work, it was beyond the scope of this work. We will make sure that this is made clear in the final version of the paper.
>
> __*Manual evaluation was conducted over only 5 articles; while there may have been ~70 sentences evaluated, this does not give a particularly good measure for overall performance.*__
>
> Whilst we agree that a larger sample size would have been desirable, this was an unfortunate consequence of the costly nature of our human evaluation, which already demanded a significant cognitive effort and time commitment from our expert evaluators (primarily due to the fact each sentence had to be manually fact-checked against the input article). Again, we will endeavour to make this more clear in the final version. We would also like to highlight that the sample size is taken into account during the statistical significance testing carried out on the evaluation results, which demonstrate evidence of significant improvement (specifically, for readability) despite the relatively small sample size.
>
> __*Does the external knowledge graph decrease the abstractiveness of the summaries (target and generated)? In some sense adding the external vocabulary may be making the task more extractive than abstractive, which would be a contribution/finding worth including.*__
>
> We thank the reviewer for raising this interesting point - to investigate it, we have rerun our evaluation with the addition of an “abstractiveness” measure (equal to the percentage of novel unigrams in the generated summaries) - the results for this are included below:
>
> Longformer (basline): 22.95%;
> text-aug: 20.99%;
> doc-enhance: 20.00%;
> decoder-attn: 24.46%;
>
> Interestingly, the results show that both the ”text augmentation” and “document enhancement” methods do slightly decrease the abstractiveness of the base model (as suggested by the reviewer), but the “decoder attention” method actually slightly increases it. We will make sure to include this finding in the final version of the paper.
>
>
> __*What was the global attention setting?*__
>
> We kept the global attention setting of Longformer to its default summarisation setting, as defined by the [HuggingFace](https://huggingface.co/docs/transformers/model_doc/led) (i.e., the first <s> token). Note that all code for our models will be made available in the final version.
>
> __*Why is it surprising that all methods of knowledge graph integration performed comparably? This is the same information, allowed to interact fairly richly with all the document tokens, and relatively minor differences in model architecture.*__
>
> In the case of the text augmentation method, we felt that the performance similarity was interesting  as this method is actually exposed to significantly less graph information than the other two GAT-based methods (i.e., only the concept definitions and semantic types of the UMLS concepts mentioned in the article abstract  due to model input constraints). Although we agree that (for the reasons you have given) the performance similarity in the case of the GAT-based methods might not necessarily be surprising, we think that it is an interesting and important comparison to make due to the fact that each of these methods targets a distinct component of the model architecture for knowledge integration (i.e., the encoder and the decoder).

---

### Official Review · Reviewer_sfG5 · 2023-08-05

**Soundness:** 4

**Excitement:**

4: Strong: This paper deepens the understanding of some phenomenon or lowers the barriers to an existing research direction.

**Paper Topic And Main Contributions:**

The paper describes and compare three methods to improve the lay summary of biomedical scientific articles by exploiting article-specific knowledge graphs that contains information on relevant biomedical concepts described in the articles.
The article is well written, with examples of the adopted Knowledge Graphs, a well as the description of the tested summarization methods (Decoder cross-attention, Document embedding enhancement, Article text augmentation).
Finally, the tests on eLife dataset have been evaluated quantitatively (using specific-summarization metrics) and qualitatively (by human readers).

**Questions For The Authors:**

Have you also applied also NER techniques to retrieve UMLS concepts from articles?

How long your approach need to train/test the model on your hardware? Please add these details.

**Reasons To Accept:**

The paper is well written, organized, structured and full of details.
It also applies the proposed method to biomedical lay summarization for the first time.

**Reasons To Reject:**

The method is interesting, but it doesn't propose a very innovative approach, simply comparing three previous methods with slight modification to include the KGs.

**Reproducibility:**

4: Could mostly reproduce the results, but there may be some variation because of sample variance or minor variations in their interpretation of the protocol or method.

**Reviewer Confidence:**

4: Quite sure. I tried to check the important points carefully. It's unlikely, though conceivable, that I missed something that should affect my ratings.

---

> ### Author Rebuttal · Authors · 2023-08-28
>
> Thank you for your insightful comments and your recognition of our work. Please find our response to specific comments and questions below:
>
> __*The method is interesting, but it doesn't propose a very innovative approach, simply comparing three previous methods with slight modification to include the KGs.*__
>
> Whilst we respect the reviewer's perspective, we would like to highlight that none of these methods (or indeed any knowledge graph integration method) have previously been applied to the task of Lay Summarisation. In fact, as mentioned in the paper,  such methods have primarily only been applied to news article summarisation which is arguably a much less difficult task, given its significantly shorter and less abstractive reference summaries (as well as shorter input documents). Additionally, these methods have generally only been applied to internally derived knowledge graphs (i.e.,constructed from the source document itself). Therefore, combined with the various modifications made to each method and the application of these methods to external data, we would argue that the in-depth comparison of methods we offer for our specific task constitutes a significant innovation, and provides findings that will also be valuable to other types of long-form or abstractive summarisation/generation.
>
> __*Have you also applied also NER techniques to retrieve UMLS concepts from articles?*__
>
> In preliminary experiments (not included in the paper), we did try using both general-purpose and domain-specific NER models to identify biomedical concepts mentioned in articles. However, we empirically found such models to obtain much lower recall on UMLS concepts than MetaMap, which was purpose-built for this exact task. Additionally, MetaMap offered the advantage of returning UMLS concept IDs (rather than just the entity name), which allowed for the easy retrieval of other information such as the semantic type and definitions of identified concepts. Therefore, we decided to use MetaMap with an additional layer of filtering (as discussed in section 3.1 of the paper), in order to maximise the amount of UMLS-derived information we could include in our KGs. We will make sure to include these details in the  final paper version.
>
> __*How long your approach need to train/test the model on your hardware? Please add these details.*__
>
> The time taken to train each model on 2 A100 GPUs ranged from ~12 hours to 2 days depending on the specific methodology used (with the GAT-based methods - “document embedding enhancement” and “decoder cross attention” - taking  longer than the text augmentation method). We will make sure to add this information to the final version of the paper.

---

### Meta-Review · Area_Chair_pUN2 · 2023-09-11

**Recommendation:** 5

**Metareview:**

This paper investigates the use of knowledge graphs (KG) for improving lay summarization in the biomedical domain. Three strategies are proposed to integrate KG information into summarization models and experiments are conducted on the eLife dataset augmented with KG. Overall, reviewers are positive about this work. They found the paper well-written and acknowledge the novelty of applying KG integration to lay summarization. The authors  addressed all of the questions in the rebuttal, though there are remaining concerns about the size of the manual evaluation (only 5 articles).

---

### Decision · Program_Chairs · 2023-10-07

**Decision:**

Accept-Main

**Comment:**

This paper investigates the use of knowledge graphs (KG) for improving lay summarization in the biomedical domain. Three strategies are proposed to integrate KG information into summarization models and experiments are conducted on the eLife dataset augmented with KG. Overall, reviewers are positive about this work. They found the paper well-written and acknowledge the novelty of applying KG integration to lay summarization. The authors  addressed all of the questions in the rebuttal, though there are remaining concerns about the size of the manual evaluation (only 5 articles).